# ESKD Risk Prediction Model in a Multicenter Chronic Kidney Disease Cohort in China: A Derivation, Validation, and Comparison Study

**DOI:** 10.3390/jcm12041504

**Published:** 2023-02-14

**Authors:** Miao Hui, Jun Ma, Hongyu Yang, Bixia Gao, Fang Wang, Jinwei Wang, Jicheng Lv, Luxia Zhang, Li Yang, Minghui Zhao

**Affiliations:** 1Renal Division, Department of Medicine, Peking University First Hospital, Beijing 100034, China; 2Institute of Nephrology, Peking University, Beijing 100034, China; 3Key Laboratory of Renal Disease, National Health Commission of China, Beijing 100034, China; 4Key Laboratory of Chronic Kidney Disease Prevention and Treatment (Peking University), Ministry of Education, Beijing 100034, China; 5Research Units of Diagnosis and Treatment of Immune-Mediated Kidney Diseases, Chinese Academy of Medical Sciences, Beijing 100034, China; 6National Institute of Health Data Science at Peking University, Beijing 100191, China

**Keywords:** chronic kidney disease, progression, prediction model, machine learning

## Abstract

Background and objectives: In light of the growing burden of chronic kidney disease (CKD), it is of particular importance to create disease prediction models that can assist healthcare providers in identifying cases of CKD individual risk and integrate risk-based care for disease progress management. The objective of this study was to develop and validate a new pragmatic end-stage kidney disease (ESKD) risk prediction utilizing the Cox proportional hazards model (Cox) and machine learning (ML). Design, setting, participants, and measurements: The Chinese Cohort Study of Chronic Kidney Disease (C-STRIDE), a multicenter CKD cohort in China, was employed as the model’s training and testing datasets, with a split ratio of 7:3. A cohort from Peking University First Hospital (PKUFH cohort) served as the external validation dataset. The participants’ laboratory tests in those cohorts were conducted at PKUFH. We included individuals with CKD stages 1~4 at baseline. The incidence of kidney replacement therapy (KRT) was defined as the outcome. We constructed the Peking University-CKD (PKU-CKD) risk prediction model employing the Cox and ML methods, which include extreme gradient boosting (XGBoost) and survival support vector machine (SSVM). These models discriminate metrics by applying Harrell’s concordance index (Harrell’s *C*-index) and Uno’s concordance (Uno’s *C*). The calibration performance was measured by the Brier score and plots. Results: Of the 3216 C-STRIDE and 342 PKUFH participants, 411 (12.8%) and 25 (7.3%) experienced KRT with mean follow-up periods of 4.45 and 3.37 years, respectively. The features included in the PKU-CKD model were age, gender, estimated glomerular filtration rate (eGFR), urinary albumin–creatinine ratio (UACR), albumin, hemoglobin, medical history of type 2 diabetes mellitus (T2DM), and hypertension. In the test dataset, the values of the Cox model for Harrell’s *C*-index, Uno’s *C*-index, and Brier score were 0.834, 0.833, and 0.065, respectively. The XGBoost algorithm values for these metrics were 0.826, 0.825, and 0.066, respectively. The SSVM model yielded values of 0.748, 0.747, and 0.070, respectively, for the above parameters. The comparative analysis revealed no significant difference between XGBoost and Cox, in terms of Harrell’s *C*, Uno’s *C*, and the Brier score (*p* = 0.186, 0.213, and 0.41, respectively) in the test dataset. The SSVM model was significantly inferior to the previous two models (*p* < 0.001), in terms of discrimination and calibration. The validation dataset showed that XGBoost was superior to Cox, regarding Harrell’s *C*, Uno’s *C*, and the Brier score (*p* = 0.003, 0.027, and 0.032, respectively), while Cox and SSVM were almost identical concerning these three parameters (*p* = 0.102, 0.092, and 0.048, respectively). Conclusions: We developed and validated a new ESKD risk prediction model for patients with CKD, employing commonly measured indicators in clinical practice, and its overall performance was satisfactory. The conventional Cox regression and certain ML models exhibited equal accuracy in predicting the course of CKD.

## 1. Introduction

Chronic kidney disease (CKD) is a leading public health problem worldwide. The estimated prevalence of CKD is 9.1% [1] globally and 10.8% in China [2]. The disease burden has been increasing significantly because of the rise in diabetes, hypertension, and aging [2,3,4,5], and it also contributes to the increased burden of end-stage kidney disease (ESKD) requiring kidney replacement therapy, which incurs huge health costs. As a result of early asymptomatic stages of CKD and heterogeneous progression to ESKD, both healthcare providers and patients seek to predict disease prognosis for optimal risk-based individualized management.

Tangri et al. [6,7] initially developed and validated the kidney failure risk equations (KFREs) using data from two nephrology referral centers in Canada and then data from the chronic kidney disease prognosis consortium (CKD-PC), which includes millions of patients with CKD stages 3–5. For populations of Asian ancestry, the model has been externally validated in populations from Korea [8] and Singapore [9], demonstrating satisfactory performance. However, participants with early-stage CKD were sub-optimally represented in prior studies. Most participants in previous studies did not have glomerular disease, which is still the common cause of CKD in developing countries. In addition, several models (traditional statistical or ML algorithms) for ESKD prediction exist, but they are limited, partly due to sample size [10], external validation, ML model interpretation, and clinical application [11,12,13]. None of them were developed for a population with CKD stages 1–4. ML appears to make fewer assumptions and may be more accurate in predictive performance [14,15]. However, the traditional Cox regression model may lose the opportunity to identify and involve the key clinical features of CKD in the prediction model, which may be complemented by ML algorithms.

Using a multicenter CKD research cohort of patients with CKD stages 1–4 under the care of nephrologists, we used Cox and ML to develop and validate a pragmatic risk prediction model for ESKD at two and five years, based on supervised routinely available features, and we additionally compared their prediction accuracies.

## 2. Materials and Methods

### 2.1. Ethics Approval and Declaration

This study adhered to the transparent reporting of a multivariable prediction model for individual prognosis or diagnosis (TRIPOD) reporting guidelines.

### 2.2. Data Source and Study Population

#### 2.2.1. Development Cohort

This development cohort was derived from C-STRIDE [16,17], the first nationwide CKD cohort in China, along with 39 clinical centers located in 28 cities from 22 provinces. All of these clinical centers are renal departments from different hospitals. Participants who met the following criteria were eligible for enrollment: (1) aged 18–74 years and (2) specified eGFR range, according to different CKD etiologies. For glomerulonephritis (GN) patients, the eGFR should be ≥15 mL/min/1.73 m^2^. For diabetic nephropathy (DN) patients, the defining eligibility was 15 mL/min/1.73 m^2^ ≤ eGFR < 60 mL/min/1.73 m^2^ or eGFR ≥ 60 mL/min/1.73 m^2^ with “nephrotic range” proteinuria. For non-GN and non-DN patients, 15 mL/min/1.73 m^2^ ≤ eGFR < 60 mL/min/1.73 m^2^ was set for enrollment. Patients aged >70 years or without baseline eGFR and demographic values or with a follow-up time of <3 months were excluded (Figure 1). Their laboratory test data were collected in central laboratories at baseline and, along with demographics and anthropometrics, were annually collected and evaluated throughout the study. The outcomes were defined as kidney replacement therapy (KRT; maintenance dialysis or renal replacement) at three-month intervals, as ascertained by each clinical center. We censored patients without KRT events throughout the limited follow-up period due to death, dropout, or 31 December 2017, whichever came first.

#### 2.2.2. Validation Cohort

The prospective CKD cohort at PKUFH, which included CKD G1–G4 with various etiologies enrolled, was used as the external validation cohort. This cohort met the exclusion criteria of the development dataset and was, therefore, designated the validation dataset. The KRT outcome was also documented during the period of follow-up. Since the maximum follow-up length in the development set was six years, data from the patients in the validation set with a follow-up longer than six years were censored.

#### 2.2.3. Candidate Variables

The baseline visit included age, gender, resting blood pressure, and comorbidities history, including type 2 diabetes mellitus (T2DM), hypertension, and cardiovascular disease, obtained at each center. Laboratory tests were collected for each patient (hemoglobin, creatine, albumin, bicarbonate, fasting blood glucose, uric acid, blood lipids panel, and serum electrolytes). Among these items, laboratory tests, 24-h urine electrolytes, and urine ACR were measured in the central laboratory (PKUFH) to avoid variation of the testing values between laboratories. eGFR was evaluated by the CKD-EPI [18] creatinine equation. Candidate variables with missing values greater than 30% in the development dataset were excluded.

### 2.3. Data Preprocessing and Statistical Analysis

In this study, the C-STRIDE cohort was randomly divided into training and test datasets at a ratio of 7:3. The PKUFH CKD cohort was considered an independent external validation set. The training dataset was used for training survival models, while the internal test dataset and the external validation dataset were used for model evaluation.

Any feature with a missing rate greater than 30% in the development dataset was excluded. We imputed the remaining missing data using the multivariate data by chained equations (MICE) method, which can handle complex incomplete data. Moreover, to avoid the problem of information leakage, we conducted MICE imputation for the training dataset first, and then, for the test and external validation datasets, sequentially by means of the imputed training dataset. In addition, the urine ACR feature values were logarithmically transformed with a base of 10, due to a skewed distribution.

As applicable, all baseline characteristics are presented as means with standard deviations, medians with interquartile ranges (IQRs), and frequencies with percentages. An event per variable (EPV) value > 10 was used to calculate the sample size. On the basis of the candidate variables and the number of ESKD occurrences, the sample size of the development cohort was adequate. The survival station of two datasets was described using Kaplan–Meier curves. In any hypothesis test, a type I error was set at the 0.05 level.

### 2.4. Model Development and Evaluation

#### Model-Driven Feature Selection

During the building of a predictive model, feature selection is integral. We searched for the predictive feature of ESKD reported in previous studies [19,20,21]. In addition, we utilized the training set to identify potential features in conjunction with the Cox model stepwise selection (both directions, Akaike information criterion), as well as the least absolute shrinkage and selection operator (Cox–LASSO) approaches, to find the optimal model via cross-validation. Partial likelihood deviance within the acceptable ranges was determined the lambda. Expert knowledge and clinical cost-effectiveness were integrated to determine the final features.

### 2.5. Model Training

During model development, we investigated the Cox regression model and two ML models, namely XGBoost [22] and SSVM [23]. XGBoost is a gradient-boosting ensemble model consisting of a group of decision trees. It can deal with survival prediction tasks by means of learning each tree using the survival loss function setting. As another survival machine learning model, SSVM is an extension of the conventional SVM algorithm and has also been applied in biomedical studies [24].

For each ML model, we first tuned the hyperparameters by means of the Bayesian optimization algorithm, aiming at maximizing the average of the *C*-index values of five-fold cross-validation. Then, we fit the models with the optimal hyper-parameters to the entire training dataset again to acquire the final model and then applied it to the inner test and external validation datasets for prediction. Given the feature values of a patient, the corresponding prediction results included a prognostic index (PI) and an individual survival curve, which were further used for model evaluation.

#### Model Evaluation and Comparison

We considered a series of metrics to measure model discrimination. The first and most widely used measurement was Harrell’s concordance index (Harrell’s *C*-index). Uno’s concordance (Uno’s *C*) was also included, as it is preferable for high censoring cases [25]. In addition to the global concordances, we also considered time-dependent AUC (TD-AUC) for discriminability at specific time points. In this study, we mainly focused on ESKD at two and five years.

In addition, the overall performance (or calibration) was carried out using the Brier score, which calculated the squared differences between the actual outcomes and the predictions. The Brier score ranges from 0 (prediction and results are identical) to 1 (discordant prediction). Accordingly, the calibration curves visualized the difference between the predicted and observed survival probability.

The statistical inference and hypothesis tests for Harrell’s and Uno’s *C*-index, TD-AUC, and the Brier score were based on non-parametric bootstrap resampling techniques, such as variance, 95% confidence intervals (CIs), and difference tests between models. Briefly speaking, the samples in each dataset were randomly resampled 1000 times to generate a group of bootstrap values for the statistical estimation of the metrics. Then, the CIs, z-statistics, and the corresponding *p*-values in the hypothesis testing were derived based on these bootstrap metric values [26].

### 2.6. Implementation Setup

This study was conducted using Python (version 3.8) and R (version 4.1) on a server with an Ubuntu 20.04 operating system. Data imputation, Cox regression, Cox–LASSO, XGBoost, SSVM, and hyper-parameter tuning were implemented based on the R mice (version 3.14), R survival (version 3.4), R glmnet (version 4.1), Python XGBoost (version 1.5), Python scikit-survival (version 0.17.2), and Python hyperopt (version 0.2.7) packages, respectively.

## 3. Results

### 3.1. Cohort Description

The development and external validation cohorts eventually comprised 3216 (2251 patients in the training set and 965 patients in the test set) and 342 patients, respectively (Table 1 and Figure 1). The observed incidence of the outcome ranged from 411 (12.8%) events in the C-STRIDE cohort to 25 (7.3%) events in the validation cohort. The mean time to event was 4.5 and 3.4 years in the respective cohorts. The Kaplan–Meier survival curve is shown in Appendix A. The mean age of the study population was 48 and 55 years. The mean eGFR was 52.97 and 50.83 mL/min/1.73 m^2^, and more than 70% of the patients were in CKD stages 1–3. Glomerulonephritis was the primary etiology of CKD in C-STRIDE. The baseline urinary ACR median (25th-75th percentile) was 376.40 (90.80, 911.45) mg/g and 214.37 (43.55, 1058.50) mg/g. The baseline survival rate of the training sets was 0.9969 and 0.9827 at two and five years. The other baseline characteristics of the CKD patients are presented in Table 1.

### 3.2. Feature Selection

By employing multivariable stepwise Cox analysis and Cox–LASSO regression, the characteristics that predicted ESKD were reduced further (Appendix A). Age, gender, eGFR, urine ACR, albumin, hemoglobin, medical history of T2DM, and hypertension were included in the final model.

### 3.3. Model Performance

The findings of the discrimination and calibration (Brier score) are displayed in Table 2. In the Cox model, Harrell’s *C*-indices were 0.841, 0.834, and 0.761, while the Uno’s *C*-indices were 0.807, 0.833, and 0.796 in the training, testing, and validation datasets, respectively. The XGBoost model provided similar findings for Harrell’s *C*-index, but when applied to all datasets, Uno’s *C*-index values of 0.836, 0.825, and 0.822 were more consistent in three datasets. The XGBoost method performed better than Cox, in terms of the two- and five-year TD-AUC in the validation. Compared to the preceding two models, the discrimination of the SSVM model was less satisfactory, given the Harrell’s *C*-index values of 0.748 and 0.745 in testing and validation. The Cox model and XGBoost algorithm performed similarly, in terms of the Brier scores (0.065 and 0.066), while that of the SSVM was 0.070.

The calibration curves exhibited suboptimal performance at certain risk thresholds. (Appendix A). The curves of the Cox model and SSVM revealed underestimation and overestimation, respectively, in the testing set for the high-risk population at 2 and 5 years. The calibration of XGBoost was generally centered on the 45-degree line, whereas high-risk groups were overestimated.

### 3.4. Model Comparison

A comparative analysis of the performance of the Cox, XGBoost, and SSVM models demonstrated that XGBoost performed significantly better than Cox and SSVM in the training set for both discrimination (Harrell’s *C* and Uno’s *C*) and calibration (*p* < 0.001). When used in the testing set, XGBoost performed similarly to Cox, in terms of Harrell’s *C*, Uno’s *C*, and the Brier score (*p* = 0.186, 0.213, and 0.141, respectively), but was statistically better than SSVM (*p* < 0.001). The validation set showed that XGBoost was superior to Cox, regarding Harrell’s *C*, Uno’s *C*, and the Brier scores (*p* = 0.003, 0.027, and 0.032, respectively), while Cox and SSVM were almost identical concerning these three parameters.

### 3.5. Web Application

The final PKU-CKD prognostic model was displayed via a clinical decision support system (CDSS) embedded in the hospital EHR system for further regional and prospective evaluation (Appendix A). The model’s absolute risk of KRT, shown in the interface, was calculated based on Cox algorithms. The XGBoost model results were calculated at the back-end to further evaluate the prediction accuracy. We established age, gender, eGFR, and UACR as the initial conditions in the actual operation of the model. This makes it particularly applicable for use by clinicians who make decisions in the absence of additional data. For better visualization and user experience, the output score for each patient was normalized between 0 and 100, and we presented the two- and five-year ESKD risk values, rather than the survival probability. We also introduced the impact of feature values, according to the local interpretable model-agnostic explanations (LIME) algorithm [27], which could enhance the model explanation in an application. The original equation displayed the regression coefficient of the features and baseline hazard at two and five years in the Cox model (Appendix A).

## 4. Discussion

In this study, using a multicenter cohort with a CKD stage of 1–4, mainly consisting of glomerulonephritis (57.6%), we successfully developed and externally validated a CKD model to predict the absolute risk of KRT in two- and five-year periods. In the training, testing, and validation datasets, the Cox model yielded *C*-index values of 0.807, 0.833, and 0.796, while XGBoost produced almost identical results with *C*-index values of 0.836, 0.825, and 0.822. Therefore, this risk model may aid in individualized patient management in the Chinese population. Although the Cox and XGBoost algorithms were basically equivalent in the test populations, the latter was superior in its external validation.

KFREs are the most frequently used ESKD risk prediction models in the world. They were initially developed in a Canadian population and showed high discrimination and adequate calibration, validated in 31 multinational cohorts (overall *C* statistic, 0.90; 95% CI, 0.89–0.92 at two years; *C* statistic at five years, 0.88; 95% CI, 0.86–0.90) [6]. However, most of these populations were patients with non-glomerular diseases. Cardiovascular disease or death was more common than kidney failure events in most of these cohorts. In the Chinese and Asian pacific areas, glomerular renal disease is still the leading cause of ESKD, rather than diabetes-based kidney disease as in Western countries. In the C-STRIDE cohort, where nearly 60% of the participants had glomerular disease, we validated the KFREs and found that the performances, expressed as the *C* statistic of the eight-variable equation, at two and five years were 0.79 (95% CI, 0.80–0.77) and 0.75 (95% CI, 0.76–0.74), respectively (Appendix A). Similarly, in a prior glomerulonephritis cohort study, the *C* statistic in the validation cohort was 0.72 (95% CI, 0.67–0.78) [28]. This suggests that the addition of a calibration factor or remodeling may be necessary.

The C-STRIDE study recruited participants mainly from 39 kidney disease research centers across China. Within this cohort, we developed a new equation of kidney risk prediction with better performance than the KFREs [29]. Our study was developed and validated using both the Cox and ML (SSVM and XGBoost) algorithms. We realized that the ability of the Cox and XGBoost algorithms to differentiate between patients with and without ESKD was robust, whereas the XGBoost model was slightly higher than the Cox model, in terms of validation metrics. Conversely, the Cox model’s overall ability is superior to that of SSVM; this implies that the Cox model might be comparable to or even better than some ML models for time-to-event data. However, some studies have unleashed the predictive power of machine learning far beyond traditional statistics and clinical experts, such as a study assessing dry weight in pediatric patients on chronic hemodialysis [30] and predicting the risk of incident cardiovascular [31]. The ability and feasibility of ML in predictive models are still questioned, due to the inherent overfitting [11] and “black box” characteristics in most studies [32,33,34,35]. In other words, it is not possible to understand precisely how a computation approximates a particular function. Further, higher placement in machine learning does not imply superiority. Instead, ML is deeply related to traditional statistical models, which are recognizable to most clinicians and require a combination of clinician-supervised and data-driven processes. Existing CKD prognostic models mostly include well-known risk factors for disease progression to ESKD, which is inseparable from the efforts of physicians and statistics.

The strength of this study is that a CKD prognostic model was developed using a large national multicenter cohort, employing both the Cox and XGBoost methods. The predictors of our model were regularly examined in the majority of centers, making the model practical and well-suited to routine clinical practice. However, this study has several limitations. First, our model was based on a Chinese cohort, and it still needs to be confirmed in other populations. Second, this cohort recruited patients from a kidney center, and more than half of them had glomerular disease. The mean age of this cohort was younger than that of the KFRE cohort, of which the participants with kidney disease developed this condition mainly due to diabetes or hypertension. Thus, this model may be more appropriate for tertiary specialty hospitals in developing countries than for general care. Third, this model could not account for treatment-related variables, due to a lack of data on medicines that might affect CKD prognosis.

## 5. Conclusions

In conclusion, we developed and validated the PKU-CKD prognostic risk model commonly measured in clinical practice, and the overall performance was well-discriminated and calibrated. Conventional regression and ML models may show comparable robustness in predicting the progression of CKD.

## Figures and Tables

**Figure 1 jcm-12-01504-f001:**
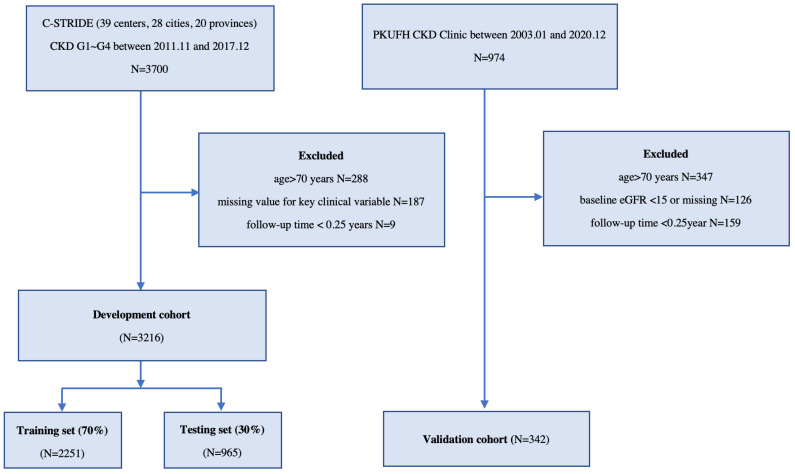
Flow chart of the screening and splitting of the study data. Abbreviations: C-STRIDE, Chinese cohort study of chronic kidney disease; PKUFH, Peking University First Hospital; CKD G1~G4, chronic kidney disease stages 1–4; *N*, sample number; eGFR, estimated glomerular filtration rate (mL/min/1.73 m^2^).

**Table 1 jcm-12-01504-t001:** Baseline characteristics of those participants with CKD in the development and validation cohorts.

Characteristics	C-STRIDE (*N* = 3216)	Missing Value	PKUFH CKD (*N* = 342)	Missing Value
Age, years	48 (13)	0	55 (11)	0
Male, *n* (%)	1909 (59.4)	0	133 (38.9)	0
Smoker, *n* (%)	1123 (34.9)	284	11 (3.2)	320
Hypertension, *n* (%)	2363 (73.5)	218	83 (24.3)	245
T2DM, *n* (%)	641 (19.9)	433	99 (28.9)	182
CVD, *n* (%)	270 (8.4)	0	7 (2.0)	269
Cause of CKD, *n* (%)		253		72
DKD	385 (12.0)		88 (25.7)	
GN	1853 (57.6)		80 (23.4)	
Other	725 (22.5)		102 (29.8)	
Systolic BP, mmHg	129.39 (17.50)	462	132.39 (18.62)	132
Diastolic BP, mmHg	80.97 (10.80)	462	77.65 (11.06)	132
ALB, g/L	38.54 (7.46)	425	42.55 (4.69)	35
HGB, mg/L	128.75 (21.87)	360	134.09 (20.18)	14
Creatinine, μmol/L	154.27 (72.17)	0	196.56 (168.08)	0
eGFR, mL/min/1.73 m^2^	52.97 (29.50)	0	50.83 (35.04)	0
>60, *n* (%) 30~59, *n* (%)	1057 (32.8) 1334 (41.5)		98 (29.0) 154 (45.6)	
15~29, *n* (%)	825 (25.7)		86 (25.4)	
UACR, mg/g	376.40 (90.80, 911.45)	329	214.37 (43.55, 1058.50)	138
<30	392 (13.6)		43 (21.1)	
30~300	889 (30.8)		64 (31.4)	
≥300	1606 (55.6)		97 (47.5)	
FBG, mmol/L	5.30 (1.69)	504	6.11 (1.56)	32
Uric acid, mmol/L	404.68 (117.17)	210	396.58 (101.87)	9
Serum phosphorus, mmol/L	1.21 (0.37)	453	1.22 (0.31)	19
Serum calcium, mmol/L	2.23 (0.20)	409	2.31 (0.15)	17
Serum potassium, mmol/L	4.44 (0.74)	446	4.43 (0.56)	8
Triglyceride, mmol/L	2.16 (1.41)	654	1.90 (1.21)	39
TC, mmol/L	5.23 (2.23)	644	4.54 (1.05)	38
HDL-C, mmol/L	1.12 (0.33)	767	1.14 (0.33)	40
LDL-C, mmol/L	2.78 (1.04)	761	2.58 (1.44)	38
KRT, *n* (%)	411 (12.8)	-	25 (7.3)	-
Follow-up time, years	4.45 (1.34)	-	3.37 (2.92)	-

Note. Values of continuous variables are presented as the mean ± SD (standard deviation) or median (interquartile ranges) according to their distribution and frequency (percentage) for categorical variables. Abbreviations: HBP, hypertension; T2DM, type 2 diabetes mellitus; CVD, cardiovascular disease; CKD, chronic kidney disease; DKD, diabetes kidney disease; GN, glomerulonephritis; ALB, albumin; HGB, hemoglobin; eGFR, estimated glomerular filtration rate; UACR, urinary albumin–creatinine ratio; FBG, fasting blood glucose; TC, total cholesterol; HDL-C, high-density lipoprotein cholesterol; LDL-C, low-density lipoprotein cholesterol; KRT, kidney replacement therapy.

**Table 2 jcm-12-01504-t002:** Discrimination and calibration of the Cox, SSVM, and XGBoost models in the training, testing, and validation sets.

**Cox Model**			
	**Training**	**Testing**	**Validation**
Harrell’s *C*	0.841 (0.811~0.871)	0.834 (0.803~0.865)	0.761 (0.696~0.824)
Uno’s *C*	0.807 (0.760~0.850)	0.833 (0.804~0.865)	0.796 (0.732~0.850)
2-yr TD-AUC	0.888 (0.847~0.927)	0.841 (0.786~0.886)	0.777 (0.689~0.858)
5-yr TD-AUC	0.696 (0.235~0.929)	0.873 (0.780~0.915)	0.905 (0.824~0.987)
Brier score	0.059 (0.051~0.068)	0.065 (0.055~0.074)	0.029 (0.022~0.037)
**XGBoost model**			
	**Training**	**Testing**	**Validation**
Harrell’s *C*	0.864 (0.836~0.891)	0.826 (0.793~0.856)	0.796 (0.738~0.852)
Uno’s *C*	0.836 (0.797~0.873)	0.825 (0.792~0.856)	0.822 (0.766~0.868)
2-yr TD-AUC	0.902 (0.863~0.938)	0.839 (0.788~0.886)	0.833 (0.762~0.900)
5-yr TD-AUC	0.754 (0.380~0.939)	0.861 (0.729~0.919)	0.912 (0.850~0.976)
Brier score	0.055 (0.047~0.064)	0.066 (0.056~0.077)	0.028 (0.021~0.035)
**SSVM model**			
	**Training**	**Testing**	**Validation**
Harrell’s *C*	0.754 (0.715~0.791)	0.748 (0.709~0.788)	0.745 (0.674~0.811)
Uno’s *C*	0.732 (0.685~0.780)	0.747 (0.702~0.790)	0.753 (0.693~0.805)
2-yr TD-AUC	0.787 (0.731~0.842)	0.751 (0.692~0.812)	0.820 (0.715~0.909)
5-yr TD-AUC	0.602 (0.187~0.879)	0.728 (0.619~0.816)	0.849 (0.791~0.896)
Brier score	0.068 (0.685~0.780)	0.070 (0.060~0.081)	0.031 (0.024~0.039)

Abbreviations: 2-yr TD-AUC, two-year time-dependent area under the curve; 5-yr TD-AUC, five-year time-dependent area under the curve.

## Data Availability

Data of this study is unavailable due to privacy or ethical restrictions.

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
