# Peer review of "ESKD Risk Prediction Model in a Multicenter Chronic Kidney Disease Cohort in China: A Derivation, Validation, and Comparison Study"

_jcm, 2023, doi:10.3390/jcm12041504_

Round 1

Reviewer 1 Report

 The study is interesting and methodologically well conducted.

However, it has several important limitations that should be addressed by the authors.

1-    The results concern only a Chinese population, and therefore require confirmations on other ethnic groups

2-    The C-STRIDE cohort is young (average age 44 years), therefore the results of the study cannot be generalized to an older population, which is moreover the one most affected by ESRD.

3-    In the C-STRIDE cohort only 12% of the population is affected by DKD, while in the western population diabetes is the main cause of ESRD.

4-    CKD is a very high CV risk condition. The study does not refer to the treatment of CKD and comorbidities. Drug treatment, especially if multifactorial, impacts the renal and cardiovascular prognosis of these patients (Cardiovasc Diabetol. 2021; 20:145. doi: 10.1186/s12933-021-01343-1. Cardiovasc Diabetol. 2022; 21: 235.  doi: 10.1186/s12933-022-01674-7). This issue should be addressed by the authors.

5-    The text contains numerous typos and requires proofreading by an English native speaker.

Author Response

Response to Reviewer 1 Comments please see the attachment

Reviewer 2 Report

The authors address a very interesting topic. Validating an ESKD prediction model risk is really very interesting.

However, reference is made to the model used in Canada, but no reference is made to the long experience of EDTA and European Nephrological Centers. The authors must remember that the important schools of nephrology must be considered.

It is advisable to read and add to the references:

1.     Locatelli F, Zoccali C; SIR SIN Study Investigators. Clinical policies on

the management of chronic kidney disease patients in Italy. Nephrol Dial

Transplant. 2008 Feb;23(2):621-6. doi: 10.1093/ndt/gfm636. Epub 2007 Nov 26.

PMID: 18039648.

2.     De Nicola L, Minutolo R, Chiodini P, Borrelli S, Zoccali C, Postorino M,

Iodice C, Nappi F, Fuiano G, Gallo C, Conte G; Italian Society of Nephrology

Study Group Target Blood pressure Levels (TABLE) in CKD. The effect of

increasing age on the prognosis of non-dialysis patients with chronic kidney

disease receiving stable nephrology care. Kidney Int. 2012 Aug;82(4):482-8. doi:

10.1038/ki.2012.174. PMID: 22622495.

      3. De Nicola L, Chiodini P, Zoccali C, Borrelli S, Cianciaruso B, Di Iorio B,

Santoro D, Giancaspro V, Abaterusso C, Gallo C, Conte G, Minutolo R; SIN-TABLE

CKD Study Group. Prognosis of CKD patients receiving outpatient nephrology care

in Italy. Clin J Am Soc Nephrol. 2011 Oct;6(10):2421-8. doi:

1.2215/CJN.01180211. Epub 2011 Aug 4. PMID: 21817127; PMCID: PMC3359552.

Furthermore, the authors have written a very important title, but it is only a "work in progress".

Authors are invited to review the paper according to the recommended revisions before publication.

Author Response

response to reviewer 2 Please see the attachment

Round 2

Reviewer 1 Report

No further comments.